# A 3.0 µm Pixels and 1.5 µm Pixels Combined Complementary Metal-Oxide Semiconductor Image Sensor for High Dynamic Range Vision beyond 106 dB [note 1]

**DOI:** 10.3390/s23218998

**Published:** 2023-11-06

**Authors:** Satoko Iida, Daisuke Kawamata, Yorito Sakano, Takaya Yamanaka, Shohei Nabeyoshi, Tomohiro Matsuura, Masahiro Toshida, Masahiro Baba, Nobuhiko Fujimori, Adarsh Basavalingappa, Sungin Han, Hidetoshi Katayama, Junichiro Azami

**Affiliations:** 1Sony Semiconductor Solutions Corporation, Atsugi-shi 243-0014, Japan; daisuke.kawamata@sony.com (D.K.); yorito.sakano@sony.com (Y.S.); takaya.yamanaka@sony.com (T.Y.); tomohiro.matsuura@sony.com (T.M.); masahiro.a.baba@sony.com (M.B.); hidetoshi.katayama@sony.com (H.K.); junichiro.azami@sony.com (J.A.); 2Sony Semiconductor Manufacturing Corporation, Kumamoto 860-8556, Japan; syohei.nabeyoshi@sony.com (S.N.); nobuhiko.fujimori@sony.com (N.F.); 3Sony Electronics Incorporated, Rochester, NY 14625, USA; adarsh.basavalingappa@sony.com (A.B.); sungin.hwang@sony.com (S.H.)

**Keywords:** automotive, HDR, LFM, motion artifact, motion blur, high definition, quadrate–square pixel, Ta–Kuchi pixel

## Abstract

We propose a new concept image sensor suitable for viewing and sensing applications. This is a report of a CMOS image sensor with a pixel architecture consisting of a 1.5 μm pixel with four-floating-diffusions-shared pixel structures and a 3.0 μm pixel with an in-pixel capacitor. These pixels are four small quadrate pixels and one big square pixel, also called quadrate–square pixels. They are arranged in a staggered pitch array. The 1.5 μm pixel pitch allows for a resolution high enough to recognize distant road signs. The 3 μm pixel with intra-pixel capacitance provides two types of signal outputs: a low-noise signal with high conversion efficiency and a highly saturated signal output, resulting in a high dynamic range (HDR). Two types of signals with long exposure times are read out from the vertical pixel, and four types of signals are read out from the horizontal pixel. In addition, two signals with short exposure times are read out again from the square pixel. A total of eight different signals are read out. This allows two rows to be read out simultaneously while reducing motion blur. This architecture achieves both an HDR of 106 dB and LED flicker mitigation (LFM), as well as being motion-artifact-free and motion-blur-less. As a result, moving subjects can be accurately recognized and detected with good color reproducibility in any lighting environment. This allows a single sensor to deliver the performance required for viewing and sensing applications.

## 1. Introduction

It is clear that image sensing is an important function in all fields today. Moving objects and obstacles must be accurately recognized and detected with high color reproducibility under all light conditions. There are many HDR methods available for CMOS image sensors. They have been shown to fall into three basic groups: the nonlinear response HDR approach, the linear response HDR approach, and the hybrid HDR approach of linear and nonlinear responses [1]. Sensing and viewing, especially in the automotive sector, must accurately perceive moving objects and obstacles and detect them with high color fidelity in all lighting conditions. For example, in order to recognize people, objects, and features even in dark places, it is necessary to sample images with high sensitivity and low noise. Also, light emitting diode (LED) traffic lights should always appear to be on in the image, even if they are actually blinking. Extending the exposure time to catch flicker tends to saturate the signal, losing luminance and color information within the pixel. To solve these problems, several HDR techniques have been proposed that extend the exposure time to capture the flicker signal but do not saturate the signal [1,2,3,4]. For example, there are types that use the overflow charge of photoelectrically converted charge in photodiodes [2,3,4,5], conversion efficiency switching and intra-pixel capacitance [6,7], adding capacitance to larger and smaller pixel sizes [8,9,10,11,12], and providing attenuation filters in small pixels without capacitance [13,14]. All of these types can achieve LFM HDR with a single exposure. However, extending the exposure time causes motion blur, which causes misrecognition. Based on these considerations, we propose an image sensor that is optimal for viewing and sensing, which combines color reproducibility and a high dynamic range while maintaining sufficient resolution and eliminating the causes of misrecognition. Until now, there has been no sensor with a dynamic range of 100 dB or higher using, for example, a pixel of 1.5 μm or less that achieves single-exposure HDR and high resolution. This is because a high capacity is required to achieve HDR, and a planar area is required to secure this capacity. Therefore, we noticed that the performance required for sensing and the performance required for viewing are different, and devised a new pixel that combines both. We realized a sensor with a dynamic range of more than 106 dB using 1.5 μm pixels.

## 2. Sensor Architecture

### 2.1. New Concept

The following features are used for sensing and viewing applications. The importance of color, resolution, and luminance depends on scene brightness. The required color and resolution information for each luminance is shown in Figure 1.

First, luminance information is necessary for low luminance below 10 candelas. At low luminance below 10 candelas, luminance information, not color information, is required to detect the presence or absence of objects. Therefore, the optimal pixels should have a larger pixel for high sensitivity and no filter and high conversion gain for low noise. Next, at high luminance of 100,000 candelas or more, luminance information, not color information, is required to detect the presence or absence of objects. Therefore, the optimal pixel should have a high saturation. Next, in medium luminance, color information and resolution are required. Therefore, the optimal pixel should have a smaller pixel and the color filters should be red, green, and clear. The reasons for this are discussed below.

The areas that are important colors for automotive sensing are indicated by the black boxes in the CIE 1976 chromaticity diagram in Figure 2. This is our original analysis. To improve sensing performance, the black-bordered areas of green, yellow, and orange require color resolution. Quantitatively expressed as a color difference (delta E), these are areas where the tolerance is very strict. In other words, the color difference (delta E) tolerance is loose for blue except for the black frame. Hence, the required color filters are defined as red, green, and clear. For this purpose, the color filter colors should be red, green, and clear.

In addition, the LED signal should not saturate even after long exposures. Therefore, we considered that the optimal pixel should have a gray filter with a small pixel size and low transmittance to reduce sensitivity. Titanium nitride was used as the material for this gray filter.

Taking advantage of the difference in information required for luminance and color, we propose the quadrate–square pixel shown in Figure 3. Incidentally, in Japanese Kanji, quadrate-Pixel is also called Ta-Pixel and square-Pixel is called Kuchi-Pixel. This is because the shape of the pixel is similar to Ta (田) and Kuchi (口). For this reason, this structure is also called Ta-Kuchi pixel.

The color filter of the quadrate pixel consists of green, red, clear, and gray, and the color filter of the square pixel consists of clear. It specializes in acquiring luminance signals by using high-sensitivity green and clear pixels in dark or low-light environments. In scenes that require color recognition in medium illumination, the signals of the red, green, and clear pixels of the pixel are used. For LED signal acquisition, low-sensitivity gray filter pixel signals that are difficult to saturate even with long exposure are used.

### 2.2. Sensor Configuration

The quadrate–square pixel arrangement method is a staggered pitch arrangement as shown in Figure 4. The number of pixels is 4.61 Mpixel for quadrate pixels and 1.15 Mpixel for square pixels. This makes it possible to obtain twice the resolution in color and luminance information compared to the 2.2 Mpixel of the usual 3 μm pitch pixel array.

Figure 5 shows the block diagram of the image sensor. A pixel array, read-out circuits (load MOS transistors, column ADCs, DAC), driver circuits (row driver, row decoder), image signal processor, and other circuits (PLL, regulator, MIPI I/F, CPU, etc.) are all mounted using a 90 nm process.

Figure 6 shows a photograph of an actual chip mounting. This is a mounting diagram of the block diagram shown in Figure 5.

A quadrate–square pixel is a combination of a 1.5 μm pixel with a four-pixel sharing configuration and a square pixel of a 3 μm pixel with an in-pixel floating capacitor (FC). Figure 7 shows a cross-sectional view of a quadrate–square pixel. The shape of the on-chip micro lens (OCL) is arranged in the same shape as the quadrate–square pixel. The light attenuation rate of gray pixels is controlled by the thickness of the gray filter and the opening width of the light-shielding layer. In a square pixel, the area of the photodiode (PD) and the FC are the same. In addition, DTI (Deep Trench Isolation) composed of an insulating film is used to electrically separate the pixels. The staggered arrangement of four small quadrate pixels and one big square pixel is a 6 μm period pixel. All of these layouts are drawn with design rules that guarantee process margins, so there is no impact on yield.

The OCL shape for one square pixel is investigated. Figure 8a shows a square OCL formed with one OCL for a square pixel and (b) shows four OCLs with the same shape as a quadrate pixel.

As shown in Figure 9, when the quantum efficiency at incident wavelengths of 530 nm and 640 nm was estimated by optical simulation, almost the same results were obtained regardless of the shape of the OCL. It is considered that this is because the light-receiving surface is sufficiently wide for the size of the light-harvesting region. In addition, the shape of the OCL is the same as that of the quadrate–square pixel b because the process stability is superior, and the quadrate OCL is applied.

### 2.3. Pixel Circuit

Figure 10 shows the pixel circuit. A quadrate pixel consists of four photodiodes, four transfer transistors (TGTs), a reset transistor (RST_T), a selection transistor (SEL_T), and a source follower amplification transistor (AMP). Four pixels are connected to one floating diffusion (FD). A square pixel consists of one photodiode, a transfer transistor (TGK), an overflow gate (OFG), a reset transistor (RST_K), a selection transistor (SEL_K), a source follower amplification transistor (AMP), and an in-pixel floating capacitor (FC).

As shown in Figure 11, the drive lines for the quadrate–square pixel are output from a single vertical scanning drive circuit as a pair with these two. The blue line is a set of drive lines for driving the Quadrate Pixel. The red line is a set of drive lines for driving the Square Pixel. Since quadrate–square pixels are staggered, the drive wiring layout is also connected in a zigzag pitch in the H direction. However, the wiring layout is designed so that the respective drive signals of quadrate–square pixels do not interfere with each other’s nodes.

### 2.4. Pixel Read-Out Method

#### 2.4.1. Square Pixel Read-Out Method

Figure 12b shows the square pixel potential diagram. The cross-section in Figure 12b shows the path of the perforation line A-B in Figure 12a. First, the PD and FC are reset to start exposure. For this purpose, TGK, FCG, and RST_K are switched to reset the charges on the PD and FC (Figure 12(b1)). During the exposure period, RST_K is turned off and FD is not put in the reset state (Figure 12(b2)). This is because the RST_K can be turned off to couple FD and TGK and lower the potential under TGK. This ensures that the saturated charge generated in the PD during the exposure period overflows the charge to the FC. When the readout period is entered, the FD is reset by turning on RST_K with TGK in the off state. RST_K is then turned off and the reset level of the square -PD is sampled (Figure 12(b3)). The difference between Figure 12(b2,b3) is that the dark current generated by the FD during the exposure period is reset in Figure 12 (b3). Then, by switching the TGK to ON, the electrons of the PD are transferred to the FD. The PD signal level is sampled. (Figure 12(b4)). Thus, in the readout period, the PD signal is read out through TGK by a CDS readout. Next, the overflow charge in the FC is read out through the FCG by a DDS readout. After FD is connected to the FC by turning FCG on, the signal level of the FC is sampled (Figure 12(b5)). Again, the RST_K is turned on and the electrical charges of FD and FC are reset and sampled as the reset level of the FC when RST_K is turned off (Figure 12(b6)). Thus, a PD signal with low noise and an FC signal with high saturation are read out.

#### 2.4.2. Quadrate–Square Pixel Read-Out Method

Figure 13 shows the read timing sequence. The signals of the square pixels and the signals of the quadrate pixels are read out continuously. The square pixel signal has two modes: a mode for reading out the charge of the photodiode with low noise due to its high conversion gain, and a mode for reading out the charge overflowing from the charge of the PD and the charge of the FC. Two rows are read out with 8AD in a 1H period by combining a long-exposure-time readout and, additionally, a short-exposure-time readout.

In the square pixel, two types of signals are read out in a single exposure. First, an exposure of the square PD and FC begins by the reset of the PD and FC. Then, the square PD reset level is sampled and, next, the PD signal level is sampled. By performing correlated double sampling (CDS) for the reset and signal level, the signal is read out. Subsequently, the signal that comes from the FC is read out by performing delta reset sampling (DRS): the FC, in which the signal level is sampled first, followed by the reset level. Because the signal charges are accumulated in FD, FD cannot be reset prior to sampling the signal level. The flaw of DRS is that kTC noise cannot be removed; however, it can be suppressed by securing the capacitance of the FC sufficiently. Subsequently, the quadrate pixels red, gray, green, and clear are sequentially read out. Finally, the charges of the square pixels accumulated for short exposure are read out again. This is for motion blur reduction, also known as motion-blur-less, and for a high dynamic range. Although it is a complicated timing control, an accumulation time setting of 8AD cycles is applied. Therefore, there is no Interference noise caused by the shutter operation of other rows during the read time, and the characteristics are not particularly affected.

## 3. Sensor Characteristics

### 3.1. Quadrate Pixel Characterristics

Figure 14 shows the output against the light intensity of the quadrate pixel.

Ideal linearity is achieved with respect to the amount of light for all colors. The minimum linear full-well capacity (FWC) is 9400 e−, and random noise (RN) is 1.4 e−.

Figure 15 shows the quantum efficiency for each wavelength of clear pixels, green pixels, red pixels, and gray pixels. Clear pixels show up to 82%.

Also, the sensitivity of the gray pixel for the LED is 550 e−/lux·s, which is 1/20 of the clear pixel sensitivity of 10,800 e−/lux·s. As a result, the dynamic range for gray pixels is 103 dB, and the illuminance saturation is 5600 cd/m^2^ after 11 ms accumulation. In addition, constant and stable quantum efficiency is obtained for wavelengths in the visible light region.

Figure 16 shows a Macbeth chart taken using this pixel with a light source of 6500 K. It exhibits high color reproducibility.

It is said that an exposure time of more than 10 ms is usually required to capture the flicker signal of the LED. From our various analyses, we estimated that the brightness of the LED signal that can distinguish the luminance of the LED from the sky is more than 4000 cd/m^2^ in the LED signal with the sky as the background. Thus, the sensitivity S required for the gray pixel is expressed as (1) with full-well capacity FWC, illuminance on image plane E_IP_, exposure time t_exp_, lens F-number F, lens transmission rate T_trns_, and luminance at saturation B.
(1)$$S<\frac{FWC}{E_{IP}{\times t}_{exp}}=\frac{FWC}{t_{exp}}\times (\frac{4\times F^{2}}{T_{trns}\times \pi \times B})$$

Assuming that the lens F-number *F* is 1.6, exposure time t_exp_ is 11 ms, luminance B is 4000 cd/m^2^, and the full-well capacity FWC is 6900 e−, the sensitivity S at the transmission rate t of 0.507 is calculated to be about 1000 e−/lx·s or less, and the sensitivity at lens transmission rate T_trns_ of 0.9 is calculated to be 550 e−/lx·s or less. As a result of prototyping with the target set as described above, the sensitivity S was 550 e−/lx·s and the full-well capacity FWC was 9400 e−. Thus, luminance at saturation B of 10,000 cd/m^2^ at a lens transmission rate T_trns_ of 0.507 and luminance at saturation B of 5600 cd/m^2^ at a lens transmission rate T_trns_ of 0.9 were achieved. Sufficient image surface saturation illuminance was achieved for a target luminance at saturation B of 4000 cd/m^2^.

Table 1 shows a comparison of the main characteristics of the color filter combinations. The important characteristics to be compared are illuminance saturation, sensitivity, and color reproducibility, and each is ranked in order of priority. The highest priority is given to illuminance saturation for LFM. This is also the most necessary as a function. Next is high sensitivity. This is needed for low-light scenes and for motion detection. Next is color reproducibility. The RCCGray and RGCB with the clear filter have the most advantage in sensitivity, which becomes important in low-light scenes. Also, the RGCGray and RGBGray with the least sensitive gray filter have the most advantage in illuminance saturation, which becomes important for LFM. The sensitivity of blue is about 1.5 times that of gray. Therefore, the saturation illuminance of B pixels is also about one-fifth that of gray pixels. In terms of color reproducibility, our own analysis shows that green, yellow, and orange color resolutions are important, and blue is less important for improving sensing performance as shown in the chromaticity diagram in Figure 2. The tolerance of the color difference (ΔE) is loose for blue, even without a blue filter. Based on these individual characteristics, the best overall color filter combination is RGCGray.

### 3.2. Square Pixel Characterristics

#### 3.2.1. Transfer Structure of Overflow Gate for PD Characteristics

In order to determine the optimum structure of the OFG, a sensor using a vertical transfer gate (VTG) and a sensor using a planar transfer gate (PTG) for the overflow transfer path were fabricated experimentally. Figure 17a shows the VTG structure and (b) shows the PTG structure. For each, the dependence of the dark current of the PD and the photo response non-uniformity (PRNU) of the FC on the applied voltage to the OFG is shown.

As shown in Figure 17a, at the oxide film interface of the VG, it is necessary to apply a lower negative bias of the OFG to strengthen the pinning and suppress the dark current. On the other hand, when the negative bias is lowered, the transfer path from the PD to the FC is not smooth and the PRNU increases. In the VTG, both the dark current and the PRNU fluctuate greatly with respect to the OFG, and there is no compatible solution.

On the other hand, as shown in Figure 17b, the PTG can form a transfer path deep from the Si surface while suppressing the pinning of the Shallow Trench Isolation (STI) or silicon surface, which causes the dark current. Since the charge can be surely overflowed without depending on the negative bias of the OFG, a compatible solution can be found.

Figure 18 shows the dependence of PD saturation charge on OFG negative bias for the VTG and PTG. Similar to the aforementioned dark current and PRNU, the amount of PD saturation charge is also determined by the potential under OFG; because VTG has a higher modulation, the decrease in PD full-well capacity is more pronounced at higher OFG applied voltages than at the PTG.

#### 3.2.2. Analysis and Verification

The ability of saturated charge overflow from the PD to FC, which is a factor for reducing the PRNU of the FC described in Section 3.2.1, is discussed. As shown in Figure 19, the theoretical value of the potential difference V_diff_ between the OFG and TGL required for saturated charge to completely flow out through the OFG side is estimated.

The outlet of saturated charge is where the potential around the PD is low. This main location is under the OFG, but the next candidate outlet is under the TGL. If almost all charge flows out through the OFG from the PD to FC, there will be no deterioration of the PRNU of the FC. However, when the PD signal is read out, since the transfer path is designed to be completely transferred under TGL, even if the gate of the TGL is negatively biased to make a high potential, a high potential may not be sufficiently obtained. Therefore, there is a component of saturated charge flowing out of the PD to the FD through the TGL. Given the above, it is verified using Equations (2) and (3):(2)$$t_{out}=t_{0}\times exp\left(\frac{V_{diff}}{kT}\right)=\frac{A^{2}}{D_{e}}\times exp\left(\frac{V_{diff}}{kT}\right)>t_{exp}$$
(3)$$D_{e}=\frac{kT}{q}\mu_{e}$$

After the potential of the PD reaches the saturation level, the time at which the generated charge at the PD flows out through the TGL to the FD is t_out_, and the exposure time is t_exp_. The diffusion coefficient of electron D_e_, the temperature T, the Boltzmann constant k, the electron mobility µ_e_, and the charge element charge q are also set. A is the distance from the PD to FD via the TGL when V_diff_ = 0. The charge that reaches FD the fastest in time is the charge generated at the PD end immediately adjacent to the TGL, and A is the shortest distance at this time. t_0_ is the average diffusion time over distance A, and Equations (2) and (3) themselves calculate the potential in the diagonally dotted area in Figure 19, which is a stricter view than the actual one. Equations (2) and (3) are established based on the current density equation, continuity equation, and diffusion coefficient. Charge generation and charge recombination are assumed to be sufficiently small. If the exposure time is longer than the outflow time, it means that the outflow to the FD through the TGL is prevented.

The worst case for the shortest outflow time is when the location of the overflow charge is closest to the TGL. This is the approximate distance where a high potential can be secured to the FD by the actual transfer gate applied voltage. In this case, the distance A is set to 0.3 µm.

Figure 20 shows the calculation results based on Equations (2) and (3). The horizontal axis is the exposure time t_exp_, and the vertical axis is the potential difference V_diff_ that can prevent charge leakage through the TGL during the exposure time. In other words, it is the potential difference when the outflow time is longer than the exposure time. Temperature dependence is also shown.

The longer the exposure time, the larger the V_diff_ because a potential difference is required to ensure that the time for charge to flow out is sufficient. In fact, the exposure time is usually set to 11 ms, indicated by the dashed line in Figure 20.

The validity of the calculated predictions shown in Figure 20 was verified by the results of actual measurements. Figure 21 shows the results of the OFG dependence of the PRNU of the FC measured at 85 °C for 11ms accumulation. The horizontal axis is the difference voltage between the actual TGL and OFG applied voltage in (a) and the potential difference between the TGL and under OFG obtained by simulation in (b).

Figure 21a,b show almost similar trends, and the potential difference at which the PRNU of the FC converges is 0.57 V from (b). This is almost equal to the 0.6 V obtained in Figure 20, indicating that this mechanism is correct.

#### 3.2.3. Transfer Structure of Overflow Gate for FC Characteristics

The optimal OFG structure for FC dark current and white spot characteristics was also examined. Figure 22 shows the dependence of FC dark current on OFG negative bias in the VTG and PTG: FC dark current is almost independent of the OFG, but the VTG’s dark current and white point are about four times worse than the PTG’s.

Figure 23 shows the temperature characteristics of the histogram of white spots in the VTG and PTG. Here, too, the VTG is about four times worse than the PTG.

The reasons for this are discussed. The temperature dependence of the dark current indicates that the activation energy is 0.54 eV, and the current is generated through generating centers with deep energy levels within the depletion layer. This indicates that the causative generating centers are distributed within the depletion layer. In the charge storage region of the FC, depletion regions exist in the Si surface oxide film, the CION sidewall, and the VTG sidewall. In other words, the depletion layer on the VTG surface is a factor; depletion of the VTG is inevitable in the charge storage type of the FC. From this point of view, the best transfer gate structure for OFGs is the PTG.

#### 3.2.4. Square Pixel Characteristics

Figure 24 shows a cross-sectional view of a portion of a square pixel having a pixel transistor.

The PD signal transfer electrode TGK is a vertical transfer gate (VTG), and the overflow electrode to the FC is a planar gate. By arranging the VTG on the TGK side, it is possible to collect the charge generated in the photoelectric conversion area efficiently, which is almost the entire area of the 3 μm square area. The OFG side is a planar type so that the saturation charge can surely overflow to the FC while minimizing the dark current generated in the PD and FC.

As shown in Figure 25, the PD full-well capacity and photo PRNU of the FC have a trade-off relationship depending on the OFG voltage.

Figure 26 shows the output against the light intensity of the square PD and square FC.

The sensitivity is 40,400 e−/lx·s, the PD full-well capacity is 13,500 e−, and the PD + FC full-well capacity is 280,000 e−. Since RN is 1.4 e−, the dynamic range is 106 dB using Equation (4). In many papers, the denominator N in the SN calculation is defined as random noise.
(4)$$Dyamic\;Range=20log(\frac{FWC}{RN})$$

Figure 27 shows the illuminance vs. signal-to-noise ratio (SNR) when combining two types of signals with 10 ms accumulation and two types of signals with 0.16 ms accumulation.

The high conversion gain signal is used in the low-illumination area, and the PD + FC signal is used in the high-illumination area. The SNR drop amount when connecting from the square PD to the square FC also maintains 30 dB at 85 °C. The dynamic range from the SNR graph is 138dB using Equation (5) when combined with a 0.16ms short exposure. For practical purposes, this is often expressed using SNR1 in the denominator:(5)$$Dyamic\;Range\;at\;SNR=20log(\frac{FWC\cdot Exposure\;ratio}{SNR1})$$

Figure 28 shows the quantum efficiency of the quadrate pixel. The quantum efficiency is 85%, which is the same as that of the quadrate pixel. The wavelength dependence is almost the same as that of the quadrate pixel. The sensitivity is high at 40,400 e−/lx·s thanks to the clear pixel.

### 3.3. Quadrate–Square Pixel Characteristics

The sensor parameters for quadrate–square pixels are summarized in Table 2.

Table 3 shows a comparison of the characteristics reported previously [6,9,13]. The well-balanced characteristics have been achieved using a CMOS image sensor with a quadrate–square pixel architecture.

### 3.4. Synthesized Image

Figure 29 shows an image of a moving object. In (a) is an image of one-shot HDR in a long exposure for LFM. Motion blur occurs during exposure. As shown in the red circle, the letters and numbers on the license plate cannot be read because of motion blur. Also, the hole in the wheel of the tire cannot be recognized. In (b) is an image of digital overlap (DOL) HDR synthesis with time-division exposure, and motion artifact occurs. Part of the image is colored green or purple, as indicated by the red circle. This occurs when combining long and short exposures in a high-contrast scene. As a result of applying the synthesis gain, when the output of the green pixel signal is higher than the pair of red and blue pixels, it becomes green, and when the output is lower, it becomes purple. In (c) is an image when using quadrate–square pixel architecture. Being motion-artifact-free and motion-blur-less are realized because it includes long exposure and short exposures with a square pixel.

Figure 30 shows the flow of motion detection applied to this work. Each pixel signal of the square is compared between the long-exposure and short-exposure signals for each pixel to determine if there is a signal difference. At this time, the square signal of short-time accumulation is multiplied by the exposure ratio gain and compared. If there is a difference, the short-time exposure signal is selected. If there is no difference, the long-time exposure signal with less noise is selected. If there is a difference, it is assumed that there is motion and a signal with a short exposure time is selected. Incidentally, the short-exposure square signal is corrected for the exposure ratio and sensitivity ratio. In other words, at this time, the square signal of short exposure contains digital noise. Therefore, the square signal of long exposure has less noise. In this way, motion detection is performed for each pixel and synthesized. In this way, this motion detection achieves both “motion-artifact-free” and “motion-blur-less” motion detection.

Figure 31 shows the road signs captured at 25 m, 35 m, and 50 m distances. Figure 31a shows an image captured using a Bayer array with a 3 µm pixel where numerical values 50 m away cannot be read. Figure 31b shows an image captured using a Bayer array with a 2.25 μm pixel. With this pitch, the numerical value 50 m ahead can be read. Figure 31c shows an image captured using a quadrate–square pixel architecture. This performance is equivalent to the 2.25 μm Bayer array.

This time, we compared it with a pixel of 2.25 µm, which is just between 3 µm and 1.5 µm in size. At least with this size of 2.25 µm, we can confirm that the quadrate–square pixel shows about the same resolution, but we will have to keep checking to see which resolution is better than the quadrate–square pixel between 1.5 µm and 2.25 µm.

As shown in Figure 32a, in principle, interpolation with a Bayer array has limits on the performance of demosaic processing. A direction detection error occurs every frame. However, as shown in Figure 32b, in this work, thanks to the array of 1.5 μm pixels, there are clear pixels in all pixels within the 3 μm pitch. Thanks to this clear pixel, there is no need for direction detection because clear has all pixels. Horizontal, vertical, and diagonal edge detection is easier than the 3 μm pitch Bayer array, and aliasing is improved. This also improves resolution. So, the quadrate–square pixel is advantageous for horizontal, vertical, and diagonal edge detection, improving aliasing and resolution.

Figure 33 shows a composite image of quadrate–square pixels at 22 ms exposure. (a) is an image in which green, red, and clear pixels of the quadrate pixel are used as color signals and clear pixels as luminance signals. It has ideal image quality without line defects due to complicated drive lines. The resolution is high enough to read numbers. Viewing image quality is achieved by high color reproducibility and high resolution. (b) is an image of the gray filter. Image quality without unevenness is obtained.

## 4. Conclusions

We have developed a new image sensor using new concept 田口pixel architecture for viewing and sensing applications with 106 dB DR and LFM that is motion-artifact-free and motion-blur-less: a CMOS image sensor with a quadrate–square pixel architecture consisting of a 1.5 μm pixel with four-floating-diffusions-shared pixel structures and a 3.0 μm pixel with an in-pixel-capacitor. The 1.5 μm pixel pitch allows for a resolution high enough to recognize distant road signs. The 3 μm pixel with intra-pixel capacitance provides two types of signal outputs: a low-noise signal with high conversion efficiency and a highly saturated signal output, resulting in a high dynamic range (HDR). For the two types of signals in the square pixel, we proposed the optimal OFG structure to balance the overflow capability of saturated charge generated in the PD to the FC and other pixel characteristics and discussed the necessary potential difference between the TGL and OFG. The application of the VTG and PTG to the OFG was examined. As a result, with the VTG, there is no solution to the pinning of the VTG interface and overflow capability, and with the PTG, overflow capability can be ensured while taking out the pinning on the Si and CION surfaces. The charge outflow time of the overflow charge is generally determined by the exposure time and the potential difference between the OFG and TGL, and we confirmed that the experimental results are consistent with the results derived from the exposure time. Therefore, the mechanism of overflow charge outflow and the required potential difference of 0.6 V at 85 °C were clarified. The dark current and white spots generated on the FC side were about four times worse in the VTG structure than in the PTG structure. This was attributed to the generated current, which was shown to be caused by depletion at the VTG interface. From this point of view, the PTG was shown to be the best choice for the OFG.

## Figures and Tables

**Figure 1 sensors-23-08998-f001:**
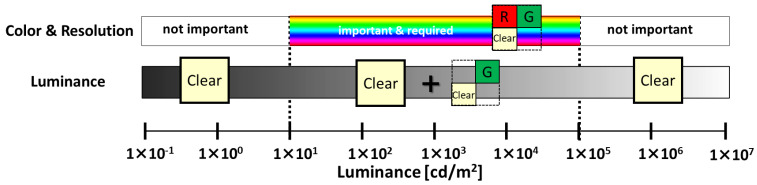
Requirements for luminance.

**Figure 2 sensors-23-08998-f002:**
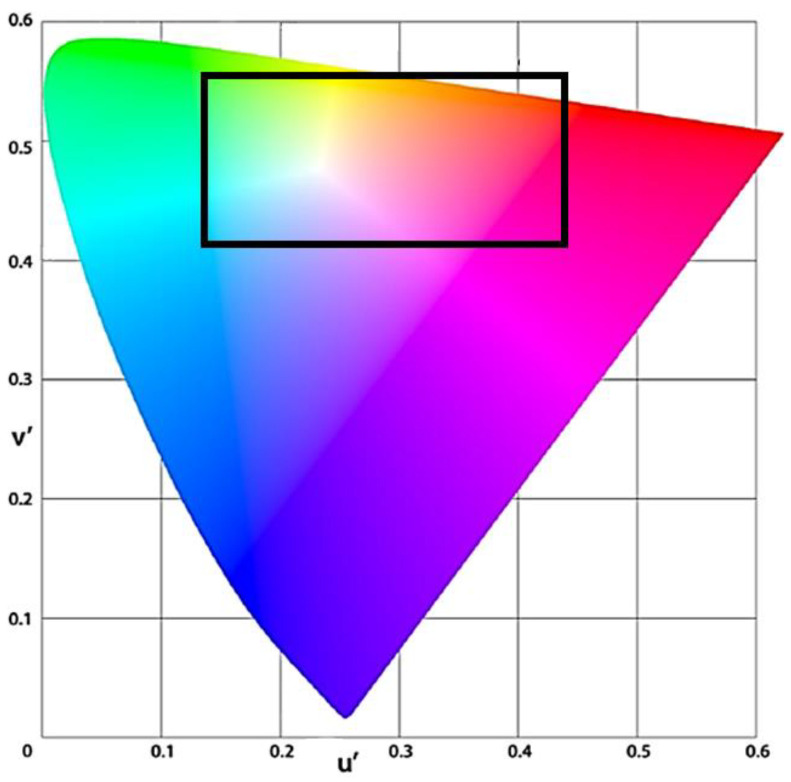
(u′, v′) chromaticity diagram, also known as the CIE 1976 UCS (uniform chromaticity scale) diagram.

**Figure 3 sensors-23-08998-f003:**
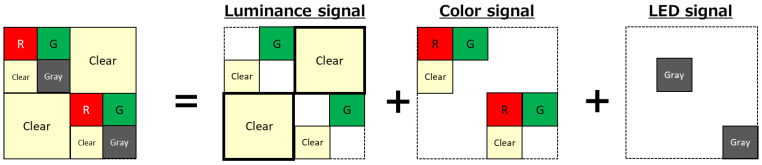
Pixel configuration.

**Figure 4 sensors-23-08998-f004:**
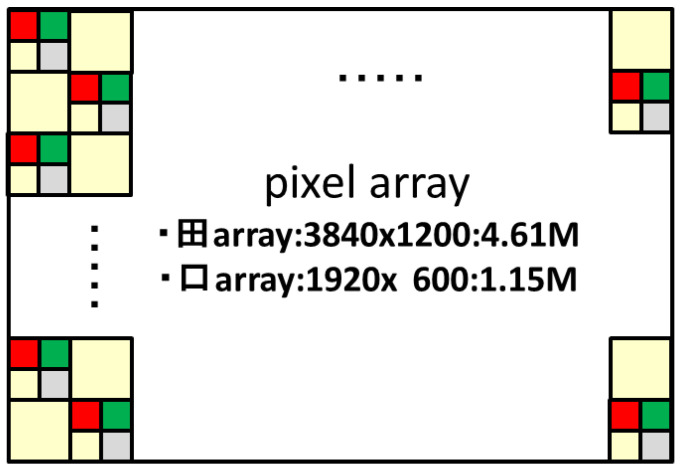
Pixel array.

**Figure 5 sensors-23-08998-f005:**
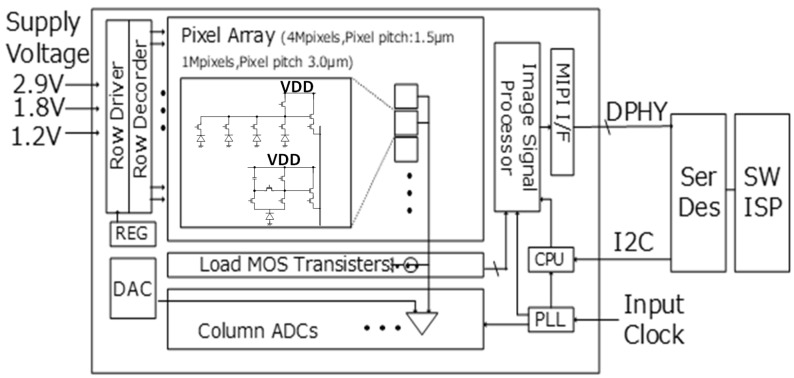
Sensor block diagram.

**Figure 6 sensors-23-08998-f006:**
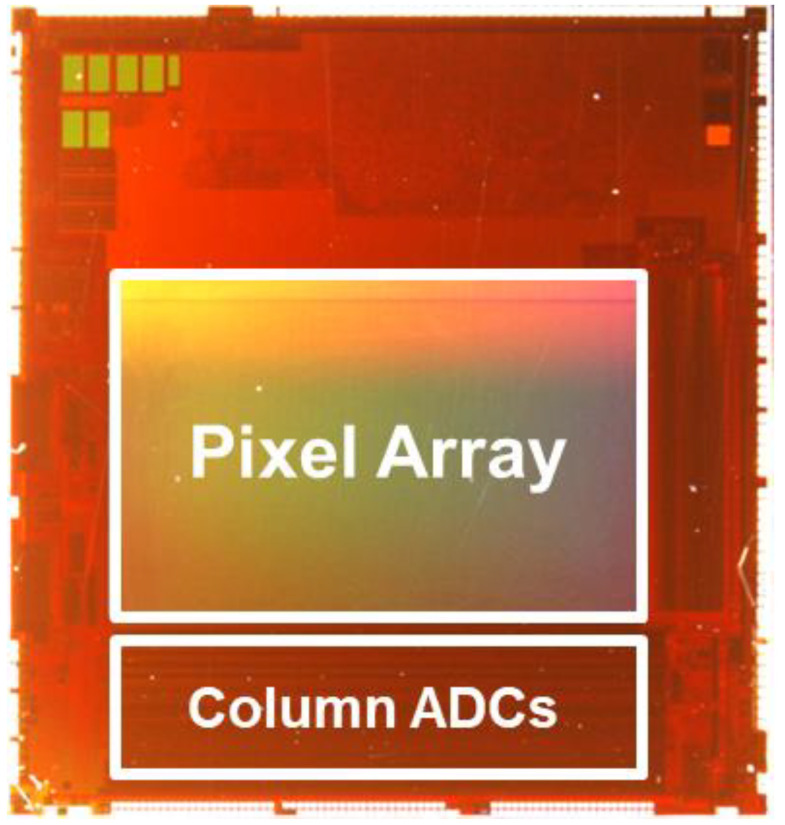
Chip implementation.

**Figure 7 sensors-23-08998-f007:**
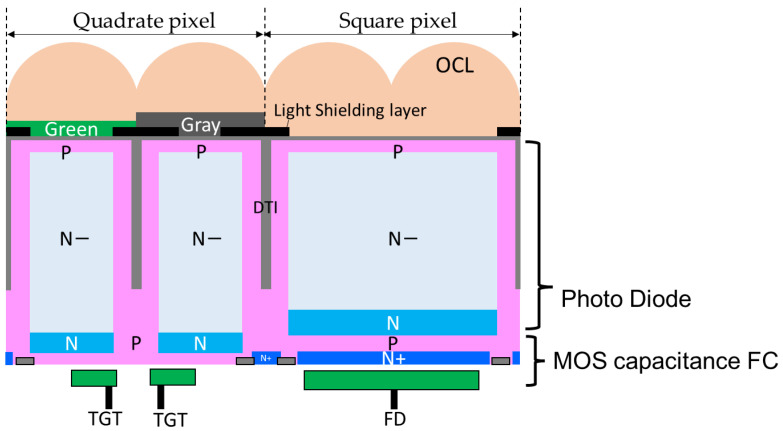
Cross-section of quadrate–square pixel.

**Figure 8 sensors-23-08998-f008:**
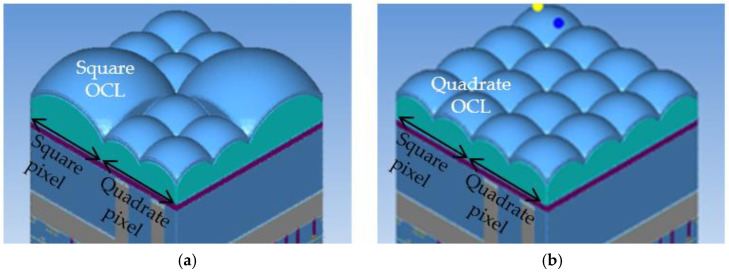
OCL shape differences. (**a**) Square OCL, (**b**) Quadrate OCL.

**Figure 9 sensors-23-08998-f009:**
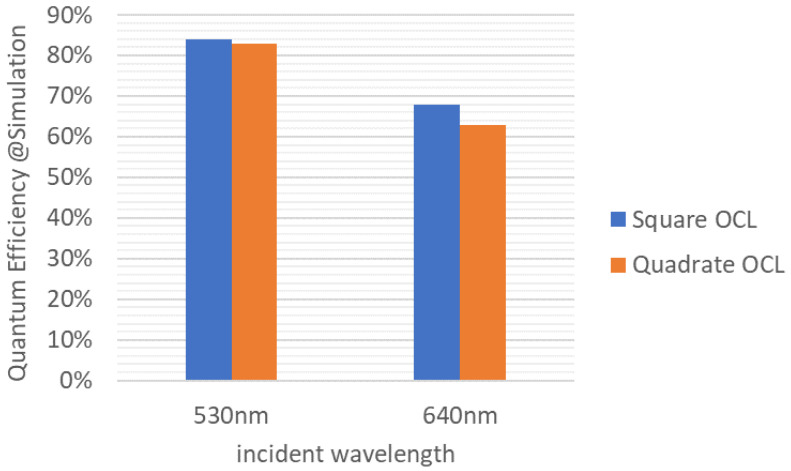
Relationship between OCL shape and Quantum Efficiency.

**Figure 10 sensors-23-08998-f010:**
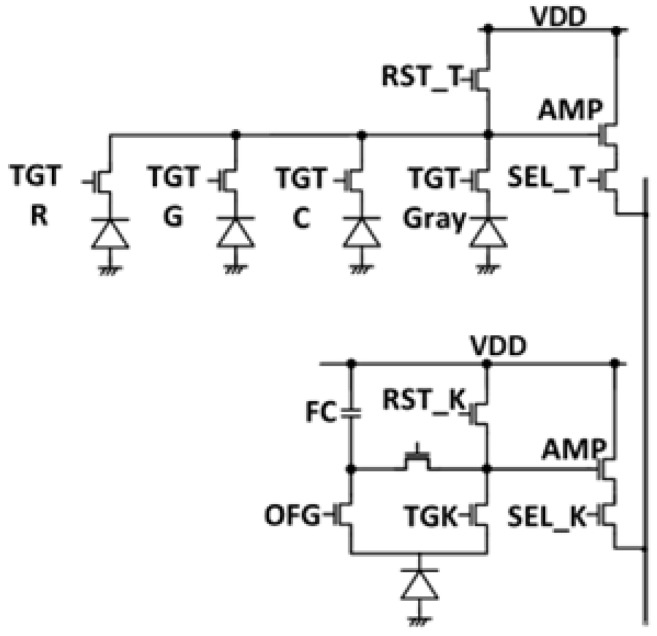
Pixel circuit.

**Figure 11 sensors-23-08998-f011:**
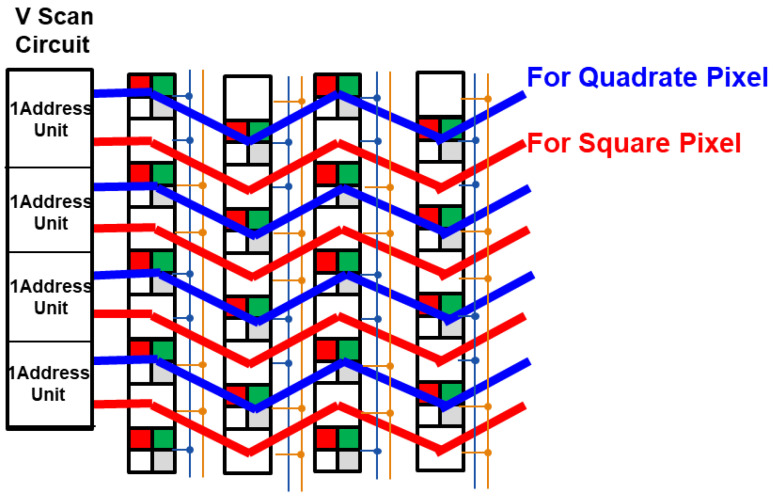
Pixel drive line layout.

**Figure 12 sensors-23-08998-f012:**
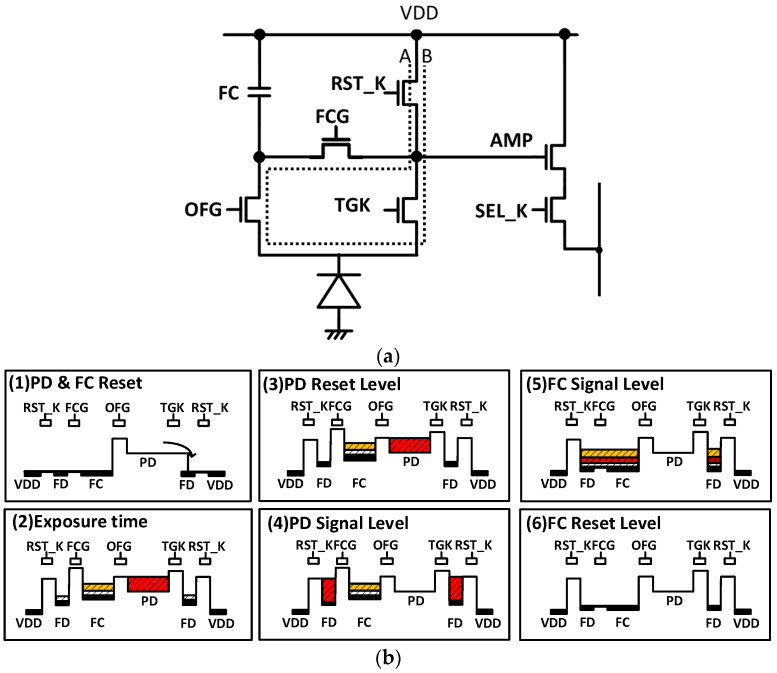
Square pixel readout. (**a**) Pixel circuit of square pixel. (**b**) Potential diagram of square pixel.

**Figure 13 sensors-23-08998-f013:**
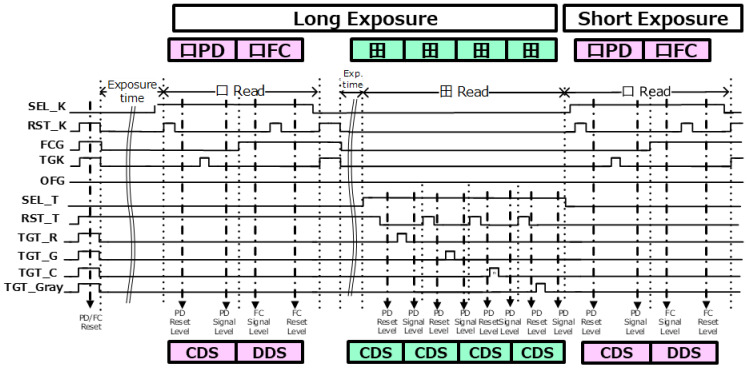
Timing sequence.

**Figure 14 sensors-23-08998-f014:**
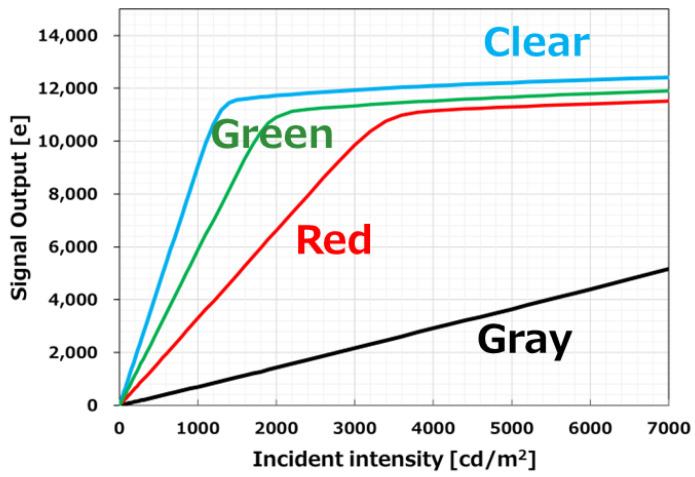
Photo response of quadrate pixels.

**Figure 15 sensors-23-08998-f015:**
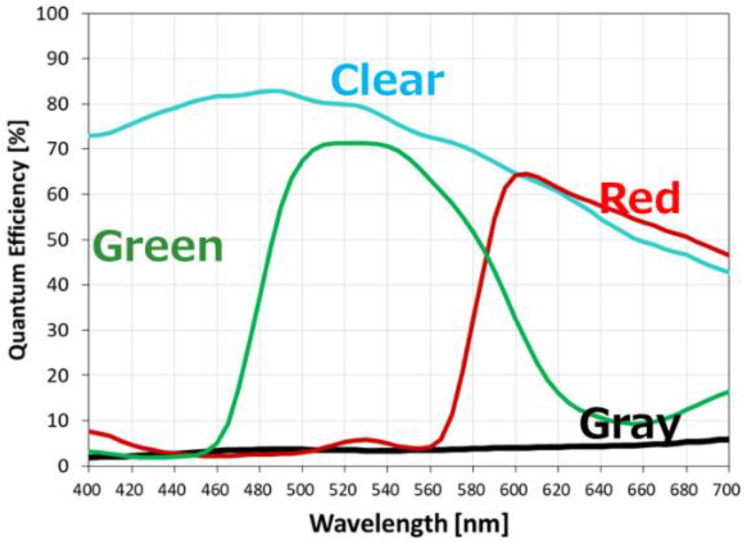
Quantum efficiency of quadrate pixel.

**Figure 16 sensors-23-08998-f016:**
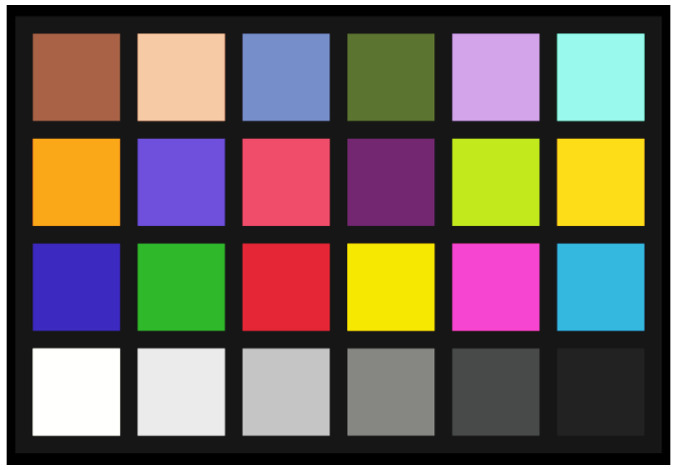
Macbeth chart at 6500 K.

**Figure 17 sensors-23-08998-f017:**
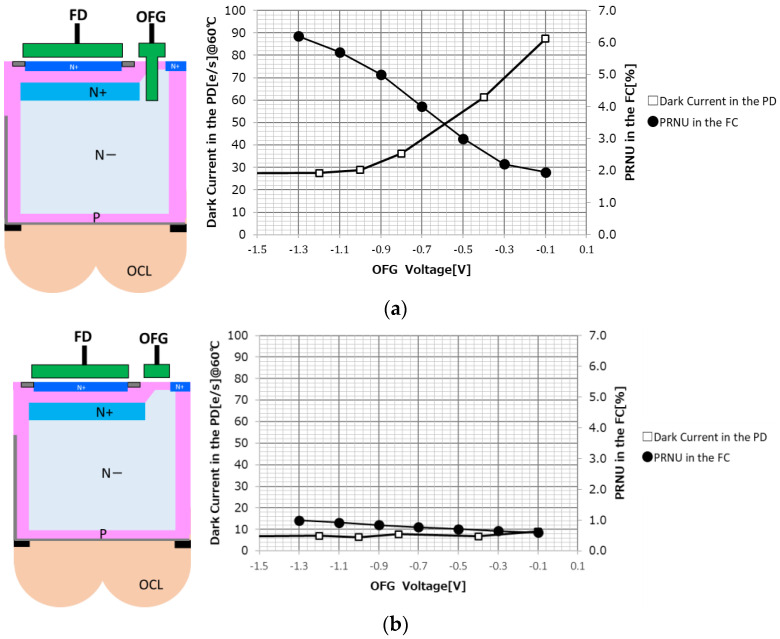
OFG dependency of dark current and PRNU. (**a**) OFG dependency with vertical transfer gate. (**b**) OFG dependency with planar transfer gate.

**Figure 18 sensors-23-08998-f018:**
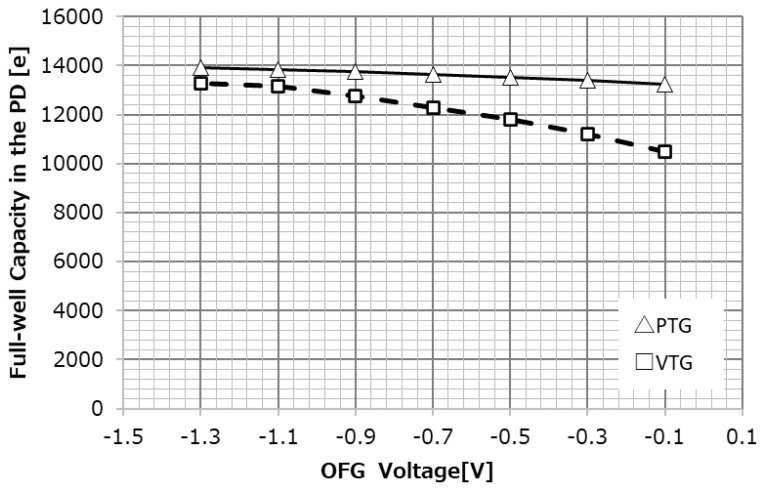
OFG dependency of full-well capacity.

**Figure 19 sensors-23-08998-f019:**
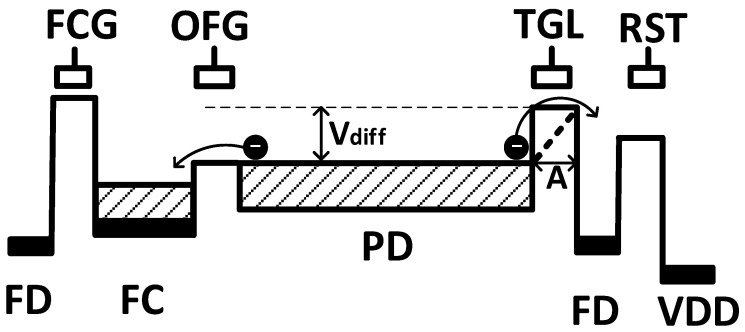
Definition of overflow charge location and potential difference.

**Figure 20 sensors-23-08998-f020:**
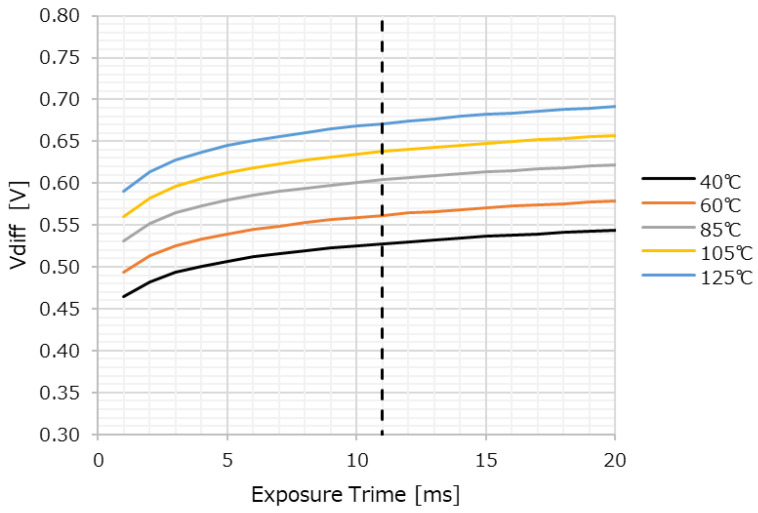
Exposure time and potential difference that can prevent overflow.

**Figure 21 sensors-23-08998-f021:**
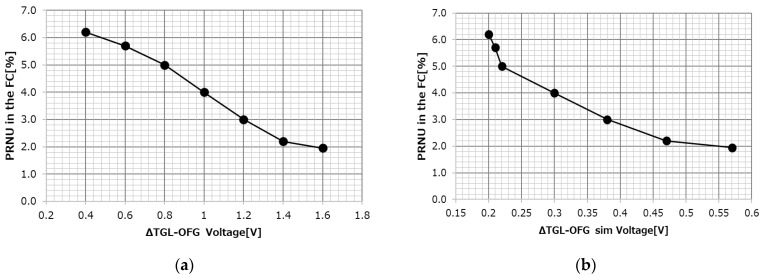
Potential difference between TGL and OFG and PRNU. (**a**) Actual setting. (**b**) Simulation.

**Figure 22 sensors-23-08998-f022:**
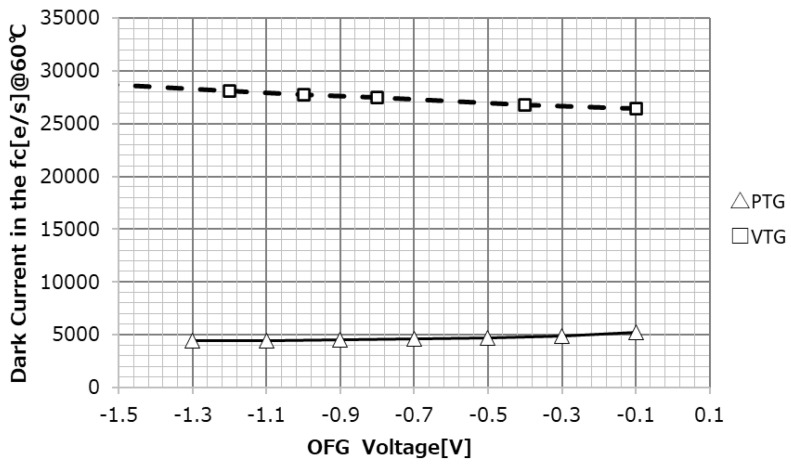
OFG dependency of dark current in the FC.

**Figure 23 sensors-23-08998-f023:**
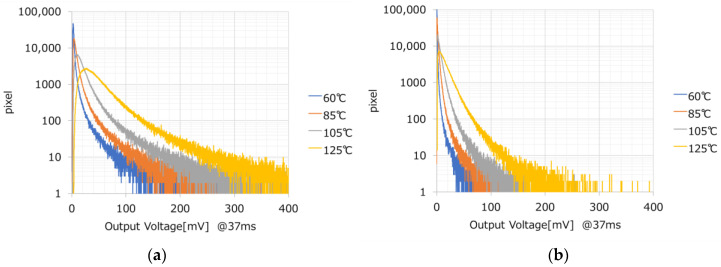
White spot of FC. (**a**) VTG structure. (**b**) PTG structure.

**Figure 24 sensors-23-08998-f024:**
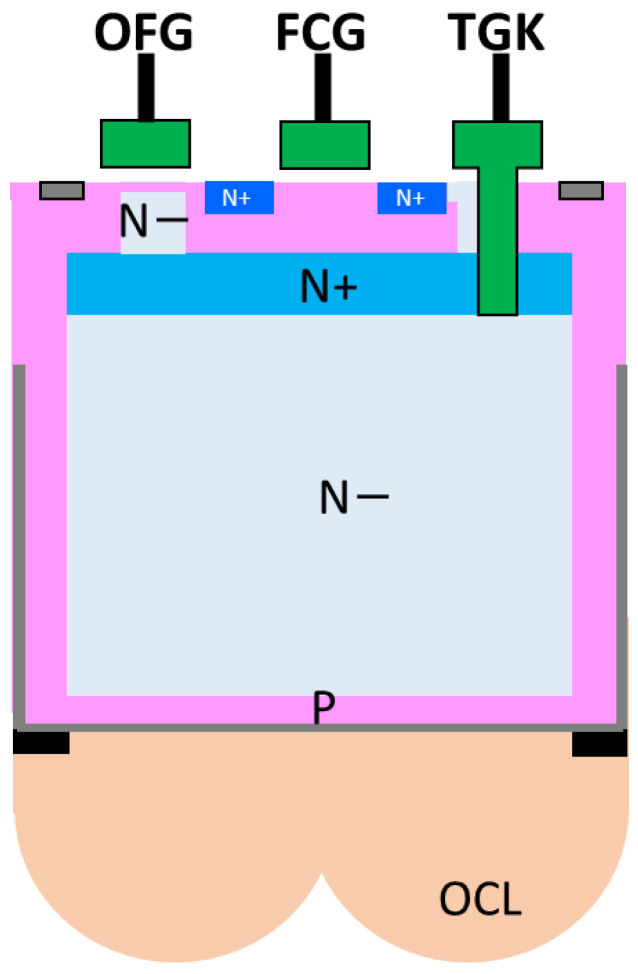
Cross-section of square pixel.

**Figure 25 sensors-23-08998-f025:**
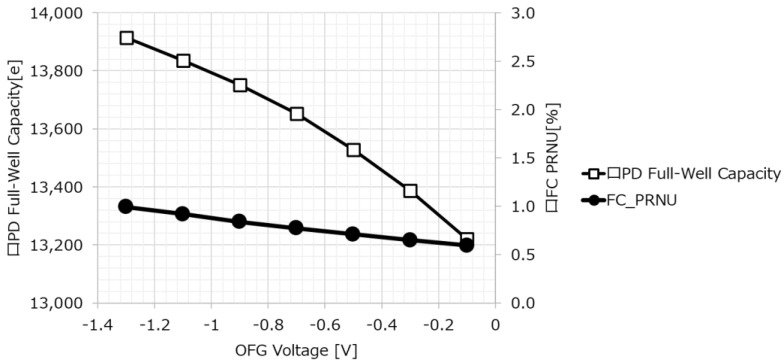
OFG dependency on FWC and PRNU.

**Figure 26 sensors-23-08998-f026:**
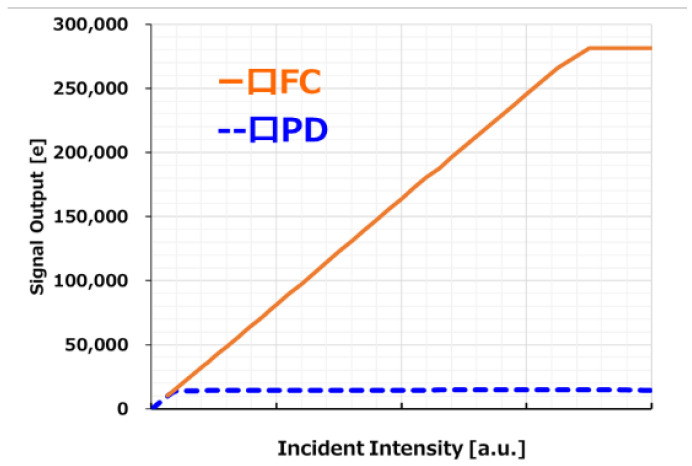
Photo response of square pixels.

**Figure 27 sensors-23-08998-f027:**
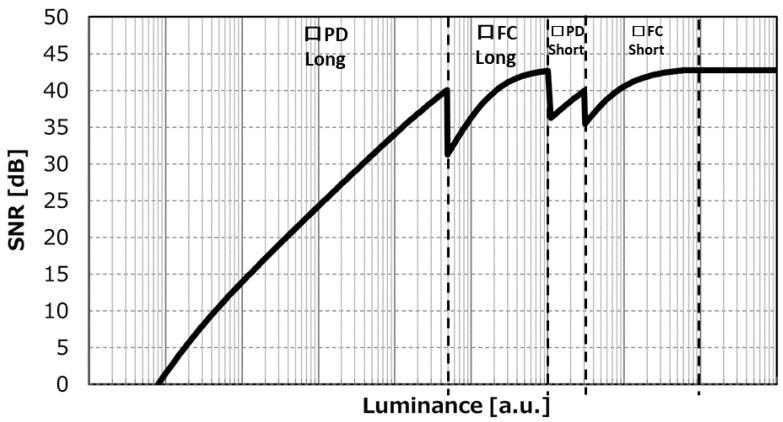
SNR curve of synthesized signal.

**Figure 28 sensors-23-08998-f028:**
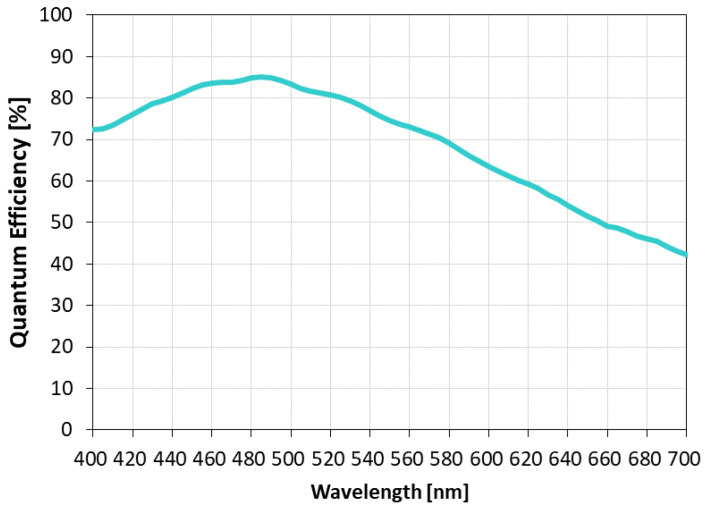
Quantum efficiency (square pixel).

**Figure 29 sensors-23-08998-f029:**
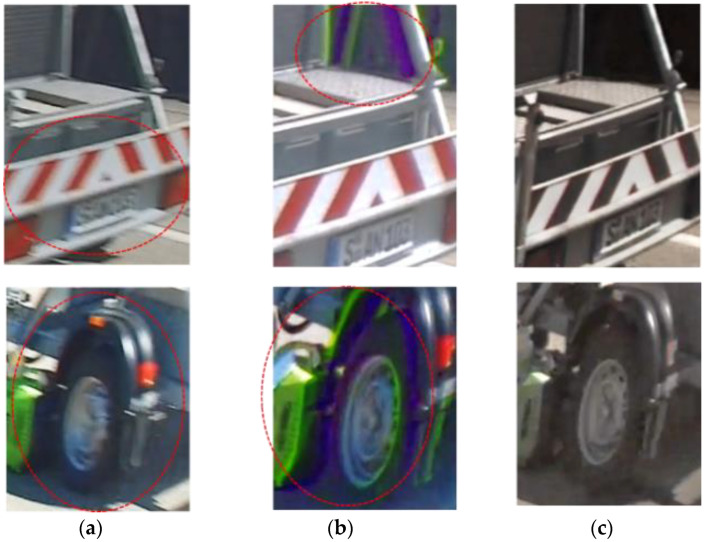
Image of a moving object. (**a**) One-shot HDR. (**b**) DOL HDR. (**c**) Quadrate–square HDR.

**Figure 30 sensors-23-08998-f030:**
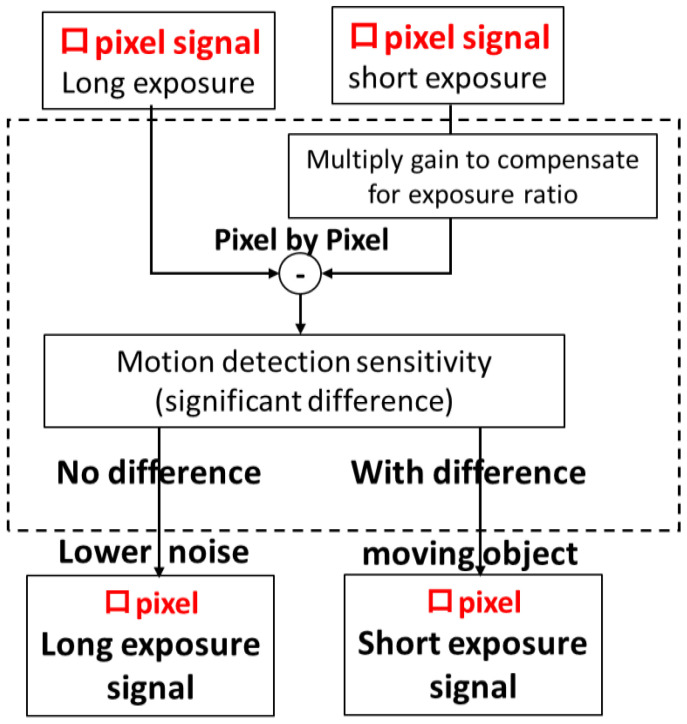
Motion detection flow.

**Figure 31 sensors-23-08998-f031:**
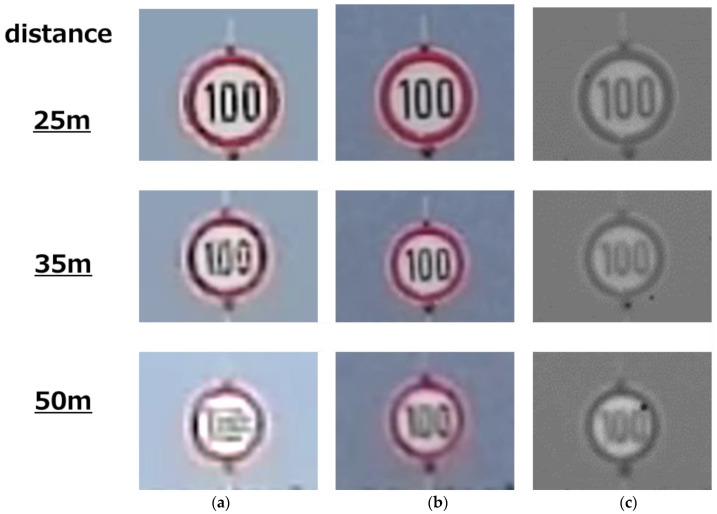
Image of a road sign. (**a**) 3 μm pixel. (**b**) 2.25 μm pixel. (**c**) Quadrate–square pixel.

**Figure 32 sensors-23-08998-f032:**
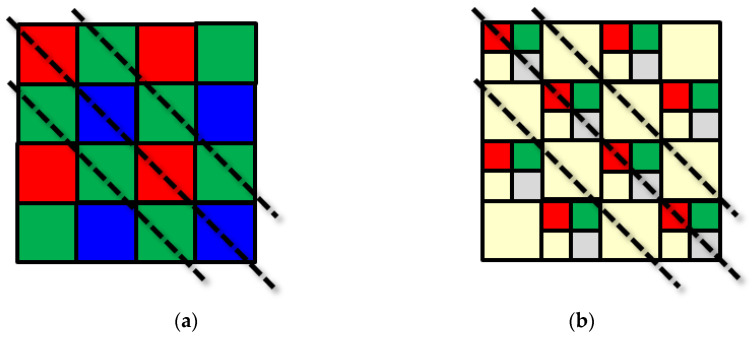
The interpolation process. (**a**) 3.0 μm Bayer array. (**b**) Quadrate–square pixel array.

**Figure 33 sensors-23-08998-f033:**
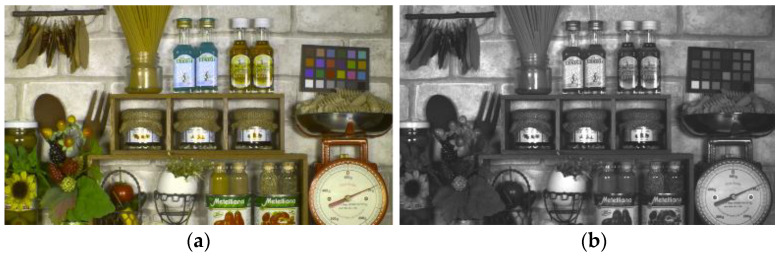
Synthesized image. (**a**) Square pixel PD + FC + Quadrate pixel RGC. (**b**) Quadrate pixel Gray image.

**Table 1 sensors-23-08998-t001:** Color filter combination comparison.

Priority	Parameter	Unit	RGCGray	RGBGray	RGGB	RGCB
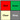	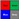	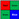	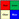
1	Illuminance saturation for LFM	cd/m^2^	5600Gray	5600Gray	1060Blue	1060Blue
2	Sensitivityfor low light scene	e−/lux·s	10,800Clear	6500Green	6500Green	10,800Clear
3	Color reproducibility	-	good	best	best	best

**Table 2 sensors-23-08998-t002:** Sensor parameters.

Parameter	Value	
	Quadrate Pixel	Square Pixel
Power supply	2.9 V/1.8 V/1.2 V	
Process technology	90 nm 4Cu 1AL CMOS BSI	
Pixel pitch	1.5 μm	3.0 μm
Pixel array	4.61 M	1.15 M
Color filter	Red, green, clear, gray	Clear
Sensitivity ratio	20 times	non
In-pixel capacitor	non	MOS
Conversion gain	81 μV/e	83 μV/e (PD), 5.7 μV/e (FC)
Random noise @RT	1.4 e−_rms_	1.4 e−_rms_
Sensitivity (3200 K with IRCF)	108,00 e−/lx·s (clear) 550 e−/lx·s (gray)	40,400 e−/lx·s
Quantum efficiency	82%	85%
Full-well capacity	9.4 K e−	1.35 K (PD),280 K e− (FC)
Real full-well capacity	188 Ke− (gray)	1.35 K (PD),280 K e− (FC)
Dynamic range (single exp.)	103 (gray) dB	106 dB

**Table 3 sensors-23-08998-t003:** Comparison of sensor characteristics.

Parameter		Unit	This Work		IEDM 2021	ISSCC 2020	IISW 2019
					[6]	[9]	[13]
Pixel pitch		μm	3	1.5	2.1	3	2.8
Color		-	Clear	RGCGray	RGGB RCCB	RGGB	RGGB
HDR Technology	Sensitivity ratio	Times	non	20	non	14.5	100
	In-pixel capacitor	-	MOS	non	MIM	MOS	non
Random noise @RT		e− rms	1.3	1.4	-	0.6	0.83
Clear or green sensitivity		e−/lx·s	40,400	10,800	N/A	38,000	24,600
Full-well capacity		e−	250 K	9.4 K	600 K	165.8 K	7.9 K
Real full-well capacity		e−	250 K	188 K	600 K	2404 K	790 K
Dynamic range (single exp.)		dB	106	103	110	132	110

## Data Availability

Not applicable.

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
