# Peer review of "A 3.0 µm Pixels and 1.5 µm Pixels Combined Complementary Metal-Oxide Semiconductor Image Sensor for High Dynamic Range Vision beyond 106 dB†"

_sensors, 2023, doi:10.3390/s23218998_

Round 1

Reviewer 1 Report

Comments and Suggestions for Authors

1.This paper has a very interesting idea and implements it into a real system. But I hope the authors should explain a few insufficient parts

2. The title of this paper is quite confusing. The phrase “for viewing and sensing applications” is too broad and quite misleading, especially the word “viewing” is very ambiguous. In other words, it may lead to a misunderstanding that LED was implemented by using CIS technology. Therefore, I hope the authors have to correct the title. For example, I recommend a title like “for High Dynamic Range Vision beyond 106dB.”

3. I think the naming of four 1.5um pixels as Ta-Pixel and one 3um pixel as Kuchi-pixel is really really Japanese style. I suggest that the pixel name should be changed into an English name which most of peoples around the world use. For example, I suggest an English expression such as “four small quadrate pixels and one big square pixel.”

4.I think it is difficult to reproduce RGB colors perfectly, because a gray filter has been inserted into Ta-pixel's filter instead of blue. What problems happens when it is normally implemented with an RGB + Gray filter? Will the HDR value be lowered? I need some explanation on this part. In other words, a clear explanation should be added into the revised version why the filter was selected as R+G+Gray+Clear, instead of normal RGGB.

5. When the authors arrange the pixels as shown in Figure 4, the yield is estimated to be poor because the pixel sizes are different. Even in memory design, if two memory cells of different sizes are placed, the yield drastically decreases. Please give some explanation on this part.

6. As shown in Figure 5, when two micro lenses used in Ta-pixel are placed on Kuchi-pixel, light refraction occurs between the micro lenses, making it difficult for Kuchi-pixel to perform properly. Please explain how the authors solve it.

7. I understand the principle of Kuchi-pixel in Figure 8. However, because it requires much more control clocks than a typical pixel, the hardware becomes complicated and noise due to the clock may occur significantly. Please explain the drawbacks of Kuchi-pixel and how to solve it. 

8. Since the authors have implemented the fabricated chips, the photos of both the fabricated chip and the measurement system should be added into the revised version. That is why the paper would be easily accepted.

Comments on the Quality of English Language

I think the naming of four 1.5um pixels as Ta-Pixel and one 3um pixel as Kuchi-pixel is really really Japanese style. I suggest that the pixel name should be changed into an English name which most of peoples around the world use. For example, I suggest an English expression such as “four small quadrate pixels and one big square pixel.”

Reviewer 2 Report

Comments and Suggestions for Authors

The submitted work mainly focus on the applications for the proposed image sensor. The results demonstrate the feasibility of the proposed method. However, it is not clear how the proposed design circuit are implemented in the real silicon.  Hence, it needs to emphasis writing the chip implementation of the proposed design.

Comments on the Quality of English Language

It might need to proofreading of the submitted work.

Reviewer 3 Report

Comments and Suggestions for Authors

The manuscript presented a novel CMOS image-sensing technique with better dynamic performance by combining 3 and 1.5-micron pixels. The results are novel and should be published in this journal. I have some minor observations summarized below.

(i) The authors should give some more information on the background and research gap in the introduction part. This will help in understanding the manuscript in a better way for the broad scientific community.

(ii) Some of the figures are not properly visible to the readers. The authors should properly revise the figures like 9, 17a.

(iii) For better understanding, the x-axis values for Figure 10 should be provided. For the current presentation, it is very hard for readers to understand the exact difference between the photoresponse nature of the different pixels.

(iv) In Figure 27, the authors claimed that their CMOS sensors have a similar performance to 2.25 micron pixels. Why the authors have only compared their sensors with 2.25-micron pixels only in this case? In other cases, they did not discuss it.

(v) The authors should provide a comparison table comparing their device performance with existing technology to properly establish the usefulness of the technique.    
